# Are Calculated Immune Markers with or Without Comorbidities Good Predictors of Colorectal Cancer Survival? The Results of a Longitudinal Study

**DOI:** 10.3390/medsci13030108

**Published:** 2025-08-01

**Authors:** Zoltan Herold, Magdolna Herold, Gyongyver Szentmartoni, Reka Szalasy, Julia Lohinszky, Aniko Somogyi, Attila Marcell Szasz, Magdolna Dank

**Affiliations:** 1Division of Oncology, Department of Internal Medicine and Oncology, Semmelweis University, 1082 Budapest, Hungary; 2Department of Internal Medicine and Hematology, Semmelweis University, 1088 Budapest, Hungary; 3Department of Medical Oncology, Semmelweis University, 1122 Budapest, Hungary; 4National Institute of Oncology, 1122 Budapest, Hungary

**Keywords:** colorectal neoplasms, longitudinal analysis, pan-immune inflammation value, prognostic nutritional index, systemic immune-inflammation index

## Abstract

Background/Objectives: Although numerous prognostic biomarkers have been proposed for colorectal cancer (CRC), their longitudinal evaluation remains limited. The aim of this study was to investigate longitudinal changes in biomarkers calculated from routinely used laboratory markers and their relationships to common chronic diseases (comorbidities). Methods: A retrospective longitudinal observational study was completed with the inclusion of 817 CRC patients and a total of 4542 measurement points. Pan-immune inflammation value (PIV), prognostic nutritional index (PNI), and systemic immune-inflammation index (SII) were calculated based on complete blood count and albumin measurement data. Results: Longitudinal data analyses confirmed the different values and slopes of the parameters tested at the different endpoints. Survivors had the lowest and most constant PIVs and SII values, and the highest and most slowly decreasing PNI values. Those patients with non-cancerous death had similar values to the previous cohort, but an increase/decrease occurred towards the death event. Patients with CRC-related death had significantly higher PIVs and SII values and significantly lower PNI values (*p* < 0.0001), and a significant increase/decrease was observed at the early observational periods. The presence of lymph node and/or distant metastases, adjuvant chemotherapy, and hypertension significantly affected PIVs and SII and/or PNI values. The changes in PIVs and SII and PNI values toward pathological values are poor prognostic signs (*p* < 0.0001). Conclusions: Each of the three calculated markers demonstrates suitability for longitudinal patient follow-up, and their pathological alterations over time serve as valuable prognostic indicators. They may also be useful to detect certain clinicopathological parameters early.

## 1. Introduction

Colorectal cancer (CRC) is the third most diagnosed cancer type for females, males and for both sexes as well, with over 1.9 million new cases and over 900,000 deaths worldwide annually, making it the second leading cause of cancer deaths according to the GLOBOCAN 2022 data [1]. Despite the large number of known cases, to date, no perfect biomarker has been identified that can be used for diagnostic and/or prognostic purposes. In recent decades, a large variety of molecular and routinely used biomarkers have been proposed, including those derived, for example, from the ratios of already routinely used laboratory markers or personalized markers. The identification of such potential biomarkers is further complicated by the fact that the number of longitudinal studies in the literature with large enough sample sizes, long enough follow-up times, and with a sufficient number of biomarker measurements is scarce [2].

In the available literature, a large number of publications have previously addressed potentially predictive biomarkers derived from blood counts and/or other routine laboratory parameters. These biomarkers have the advantages of typically requiring low budgets, being used routinely, and not requiring the development/application of new/additional techniques. Furthermore, in many cases, these combinations can indirectly indicate complex systemic and/or physiological processes and/or interrelationships between these processes. Examinations of such markers can, for example, provide valuable information about, among other things, whether the patient’s condition is worsening because of the progression of the tumor or due to other conditions [3]. While in previous decades, markers such as platelet-to-lymphocyte ratio, neutrophil-to-lymphocyte ratio, and lymphocyte-to-monocyte ratio were more frequently examined [2,4,5,6,7], definitions of the pan-immune inflammation value (PIV) and the systemic immune-inflammation index (SII), which might give further data on the complex interactions within the tumor microenvironment, have recently appeared [8,9]. Moreover, the use of prognostic nutritional index (PNI) is also experiencing a renaissance [10,11,12]. It has to be mentioned though that essentially all previous studies investigated the effect of the latter three biomarkers of CRC in only single-timepoint analyses. Either a specific timepoint of interest (e.g., pre- or postoperatively, or after the first set of chemotherapy ± biological agents, etc.) or the difference between two timepoints (also known as “delta”) was investigated.

The incidence of CRC increases with age [13]. Similarly, the incidence of common chronic diseases, such as hypertension, type 2 diabetes mellitus, asthma, chronic obstructive pulmonary disease, obesity, and various cardiovascular diseases and/or events are also increasing with older age [14]. It is quite common for CRC patients to be diagnosed not only with CRC, but also with other, incidentally found comorbidities, or to have a known, pre-existing chronic disease [15,16,17]. Comorbidities are known to significantly affect CRC survival by contributing directly or indirectly to faster disease progression. Moreover, it has also been reported that comorbidities can prevent the use of the same oncological therapies, compared to CRC patients without any comorbidities [18,19]. However, the relationship between CRC and comorbidities is not totally negative. It should be emphasized that comorbidities can play a significant role in the early detection of CRC. Patients with various chronic diseases often report new symptoms to their doctor, which can significantly help with the earlier detection of the tumor [20].

The aim of this retrospective longitudinal observational study was to investigate the effects of the PVI, SII, PNI, and their changes throughout the disease on CRC survival. The secondary aim was to study the possible associations between these markers and the various clinicopathological features, the most common incidental and/or pre-existing comorbidities, and anamnestic data.

## 2. Materials and Methods

### 2.1. Patients and Study Design

A retrospective longitudinal observational study was conducted. Patient data was obtained from the medical database of Semmelweis University, Budapest. The list of possibly eligible patients was downloaded based on the International Statistical Classification of Diseases and Related Health Problems (ICD-10) codes C18 (malignant neoplasm of colon), C19 (malignant neoplasm of rectosigmoid junction), and C20 (malignant neoplasm of rectum). After the removal of ineligible cases, a total of 817 CRC patients’ anonymous and de-identified data was retrieved. All included CRC patients attended the outpatient clinics of the Department of Internal Medicine and Hematology, Semmelweis University, Budapest, and the Department of Internal Medicine and Oncology, Semmelweis University, Budapest, between 2006 and 2020. CRC was diagnosed using colonoscopy followed by the pathological confirmation of colorectal adenocarcinoma from biopsy specimens, and further imaging studies were performed to confirm metastases. Inclusion in the study required laboratory results performed at Semmelweis University at least at the time of CRC diagnosis. Exclusion criteria included age < 18 years; any previous malignancies; a histological CRC type other than adenocarcinoma; emergency cases such as perforation and/or obstruction due to CRC; known hematologic, inflammatory bowel, systemic autoimmune, mental, hepatic (other than Gilbert’s syndrome), acute infectious, and/or inadequately controlled thyroid diseases; known renal dysfunction; the usage of systemic corticosteroids 90 days prior to the baseline visit date; erythropoiesis-stimulating agents; and patients with an Eastern Cooperative Oncology Group (ECOG) performance status > 2 (Figure 1). This study was approved by the Regional and Institutional Committee of Science and Research Ethics, Semmelweis University (SE TUKEB 21-14/1994, approval date of latest modification: 23 February 2021). Handling of patient data was performed in accordance with the General Data Protection Regulation issued by the European Union, and in concordance with the WMA Declaration of Helsinki.

Patient visits to analyze were selected randomly. A total of 4542 visits (5.56 ± 3.93 visits/patient as average ± standard deviation) for the 817 patients were processed (Figure 1). PIVs and SII values could be calculated for 3157 of these 4542 visits (number of patients: 771; average visits: 4.09 ± 3.35), while PNIs were calculated for 1610 of the 4542 visits (number of patients: 591; average visits: 2.72 ± 2.46). Patients with a surgically resectable tumor had visits at the time of tumor diagnosis prior to any oncological procedures, at least 4–6 weeks after the primary tumor removal surgery and just before the first oncological treatment, and every 4–6 months thereafter. Patient visits with an irresectable tumor were recorded for every 4–6 months following the initial laboratory measurement performed at the time of CRC diagnosis. If a patient had no laboratory measurement(s) within the optimal window, the next measurement was recorded as close as possible. Laboratory measurements around recent blood transfusion(s) were omitted and another random visit was selected as close as possible.

### 2.2. Clinicopathological and Laboratory Data Measurements

Anamnestic data including disease history was collected at the time of the baseline visit. For appendicitis and appendectomy, and cholelithiasis and cholecystectomy, the surgery and condition events were combined, respectively. This had to be performed as a technical step because in several cases, the disease(s) could be managed conservatively, and in many cases, cholelithiasis was only detected during the preoperative imaging studies. Fasting blood measurement data, including complete blood count parameters, was recorded for every visit. All laboratory measurements were completed at the laboratories of Semmelweis University, Budapest. Staging was given by histopathological examination of surgical specimens and imaging studies; the American Joint Committee on Cancer (8th edition) grouping was used [21]. The location of CRC was described as rectal cancer, right-sided, left-sided, and at multiple sites, as described previously [22]. Chemotherapy was recorded as “adjuvant” if no distant metastasis by imaging was detected, and “palliative” if metastasis was present. Furthermore, the “palliative” group was further divided into “First-line”, “Second-line”, and “Third-line and above”. The usage of biological agents—such as bevacizumab or cetuximab—and oral regorafenib/trifluridine–tipiracil was recorded as a binary variable. Survival times were calculated from the time of CRC diagnosis until the patient’s death or until the termination of data collection (31 December 2024). Both overall survival (OS) and disease-specific survival (DSS) were calculated. In the case of the latter, tumor-related and -unrelated deaths were recorded as separate events. Patients alive at the end of the observational period were right-censored. The PIVs and SII and PNI values were calculated as described previously (Equations (1)–(3)) [8,9,23]. Moreover, PNI was categorized, based on its values, as normal, mild, moderate, and severe malnutrition if the values were ≥50, 50–45, 45–40, and <40, respectively, as determined previously [23,24]. Based on patients’ PNI values, the PNI grouping varied dynamically from visit to visit (if needed) and was not fixed per patient.(1)$$PIV=\frac{neutrophyl\;count\times platelet\;count\times monocyte\;count}{lymphocyte\;count}$$(2)$$SII=\frac{platelet\;count\times neutrophyl\;count}{lypmphocyte\;count}$$(3)$$PNI=serum\;albumin\left(in\;g/L\right)+0.005\times lymphocyte\;count(in\;/mm^{3})$$

### 2.3. Statistical Analysis

Statistical analysis was performed within the R for Windows version 4.5.0 environment (R Foundation for Statistical Computing, 2025, Vienna, Austria). Natural-cubic-spline-adjusted random intercept linear mixed effect models were constructed to determine parameter changes with the course of the disease [R packages “nlme” (version 3.1-168) and “splines” (version 4.5.0)]. Two types of survival models were used. Simple competing risk models were used to investigate cumulative incidences, while for survival data associated with longitudinal data, competing risk survival models with a time-dependent covariate were applied (R package “survival”, version 3.8-3). *p* < 0.05 was considered statistically significant. Continuous data was reported as mean ± standard deviation, while the number of occurrences and their percentages in parentheses characterized the frequency data. Survival data was presented as the hazard ratio (HR) and its 95% confidence interval (95%CI). Figures were created using the R packages “survminer” (version 0.5.0) and “lattice” (version 0.22-7).

## 3. Results

As detailed in Methods, a total of 771 and 591 of the 817 patients with 3157 and 1610 individual visits had data available on their PIV/SII and PNI, respectively. The average age of the patients at the time of diagnosis was 65 ± 11 years, and there were no sex differences. There was a slight male dominance (56% vs. 44%). Since the study subjects’ comorbidities and relevant medical history were an important part of the study, recording the presence of the following comorbidities was of paramount importance: type 2 diabetes mellitus; thyroid diseases (e.g., Hashimoto’s hypothyroidism); minor and major cardiovascular (CV) diseases, such as hypertension, myocardial infarction, and stroke; and the presence/history of appendicitis and/or cholelithiasis with or without appendectomy and/or cholecystectomy. The complete list of anamnestic data of the whole, PIV/SII, and PNI cohorts can be seen in Table 1.

### 3.1. Longitudinal Data Analysis Results

For the 771 and 591 patients with PIV/SII and PNI data available, an average of 4.09 ± 3.35 and 2.72 ± 2.46 visits per patient with a mean follow-up time of 75.25 ± 56.40 months and 75.81 ± 56.68 months were found, respectively. First, it was investigated whether the PIVs and SII, and PNI values differed between those patients who were (1.) alive, (2.) dead due to CRC-related reasons, or (3.) dead due to non-cancerous reasons at the end of the observation period. Both the PIVs and SII values were constant in the surviving patients, while the PNI values slowly decreased. CRC-related death was associated with constant, significantly higher PIVs (*p* < 0.0001) and SII (*p* < 0.0001) values, and significantly lower PNI (*p* < 0.0001) values. The PIVs (*p* = 0.5487) and SII (*p* = 0.5932) and PNI (*p* = 0.1306) values of those patients who also died, but not due to CRC, were more similar to those of the survivors, showing no significant changes. It should be highlighted, however, that a significant slope of change in the values of the indices was observed in both deceased cohorts. While this increase occurred more often in the first half of the study in patients who died of CRC (PIV: *p* = 0.0277; SII: *p* = 0.0179; PNI: *p* = 0.0228), it was more pronounced in the second half of the study in patients who died of other causes (PIV and SII: statistically not significant; PNI: *p* = 0.0068; Figure 2). To gather more information about this phenomenon, the cumulative incidence of the two endpoints was investigated using competing risk survival models. The results of the latter model were in line with the former observation: death due to CRC occurred more often in the earlier stages of the study, while other deaths occurred more frequently in the later stages (Figure 3).

The secondary question(s) of our study was whether the various comorbidities and/or clinicopathological parameters affected PIVs and SII and/or PNI values (Table 2). PIVs were significantly lower in females (*p* = 0.0286; Figure 4A). Stage IV CRC (at the time of diagnosis) caused a constant significant increase in PIVs (*p* < 0.0001) and SII values (*p* < 0.0001) and a significant decrease in PNI (*p* < 0.0001) values throughout the observation period, compared to those study participants with stage I-III disease. Both regional lymph node and synchronous distant metastases significantly affected the indices (*p* < 0.001). In the case of metachronous metastases, SII and PNI values were significantly higher (*p* = 0.0396) and lower (*p* = 0.0191), respectively, and only a marginal increase in PIVs (*p* = 0.0911) was observed. SII values were significantly higher in the case of rectal tumor (*p* = 0.0147). The various anti-cancer therapies had the following effects: Those patients receiving adjuvant chemotherapy had significantly better PIVs and SII and PNI values (*p* < 0.0001) compared to those receiving palliative chemotherapy or no chemotherapy at all. The use of biological agents was also associated with higher PIVs and SII values (Table 2).

We examined the impact of comorbidities by analyzing each comorbidity separately and in combination as well. The following combinations were investigated: major CV events, all CV diseases excluding hypertension, and any comorbidities versus no comorbidities. For PIVs and SII values, no differences between the diseased and non-diseased groups could be demonstrated for either individual comorbidities or combinations. In contrast, the PNI was significantly worsened by the co-existence of hypertension (*p* = 0.0076; Figure 4B) and the occurrence of transient ischemic attack/lacunar stroke (*p* = 0.0229) and/or (pulmonary) embolism (*p* = 0.0292) in the patient’s medical history. A reduced PNI value was observed with older age (*p* < 0.0001) and when comparing the diseased and non-diseased groups (*p* = 0.0005; Table 2).

### 3.2. Survival Analysis Results

Of the 771 patients for whom PIV/SII data was available, 282, 363, and 126 patients were alive at the end of the follow-up period, died from CRC, and died from other causes, respectively. Similar rates were found for the 591 patients with PNI data. Here 211, 285, and 95 subjects were alive, died due to CRC, and died due to other causes, respectively. First, the univariate effects of PIV, SII, and PNI changes over patients’ DSS were investigated. Increasing PIVs (HR: 1.00023; 95% CI: 1.00017–1.00029; *p* < 0.0001) and SII values (HR: 1.00019; 95% CI: 1.00015–1.00023; *p* < 0.0001; Figure 5B) and decreasing PNI (HR: 0.8797; 95% CI: 0.8543–0.9059; *p* < 0.0001) values are poor prognostic markers of shorter survival times. Moreover, the PNI data was analyzed by using the PNI category grouping. Compared to those patients who had normal PNI values, those with mild, moderate, and severe malnutrition had an increased hazard ratio of 4.7756 (95% CI: 2.4314–9.3800; *p* < 0.0001), 9.9865 (95% CI: 4.7315–21.0776; *p* < 0.0001), and 57.3284 (95% CI: 30.2639–108.5961; *p* < 0.0001), respectively (Figure 5A). The same observations were found for OS, with HR and 95% CI values changing only slightly.

The DSS of patients was investigated in a multivariate setting as well. As the traditional factors for CRC significantly reduced any impact of comorbidities on survival, two sets of multivariate survival models were created. First, PIV, SII, and PNI were combined with sex and the various comorbidities. Each of the three study targets was shown to have a significant effect on survival. Of the parameters tested in the models, hypertension had a more significant effect, as did the gender of the patients in the SII model (Table 3). Second, a combination of CRC-specific parameters and the presence of comorbidities as a dummy variable were examined. As an expected result, higher disease stages and older age were significant predictors of shorter survival times. The same was true for higher PIVs and SII values and lower PVI values; all three were associated with worse patient survival. Moreover, the location of the primary tumor and the presence of any comorbidities were also poor prognostic signs in the SII and PNI models, respectively (Table 4).

## 4. Discussion

Inflammation and immune response play crucial roles in the pathology, development, progression, and metastatic spread of CRC [25,26]. The applicability of several immune and/or inflammatory biomarkers has been investigated in recent years/decades. These biomarkers are often selected from among parameters already used in routine diagnostics, but their combination/ratios are used. By using such indices, one can generate specific ratios whose shift in one direction or the other can be an indirect pathological indicator of certain physiological processes., e.g., the lymphocyte-to-monocyte ratio can be used to gather information on whether the patient’s condition is worsening because of the progression of the tumor or due to other conditions [3]. Even though these markers can provide indirect information only about the tumorous disease(s) and their relationship between cancer and inflammation and/or immunity, they have been shown to be very good predictors over the past decades and have been used in a wide range of studies. But there are still blank spots in our knowledge. One of these is that well-detailed longitudinal analyses about these biomarkers are scarce. Most of the complete blood count index-based biomarkers have been shown to have a distinct prognostic value at diagnosis, before or after surgery, before or after chemo, before or after immunotherapy, etc. In fact, they often even show very strong associations with other clinicopathological markers. However, the accuracy of these predictions is somewhat biased, as the measurements and conclusions are usually taken from a single timepoint only or are based on “delta” values. To overcome this bias, longitudinal analyses are suggested. For example, in a previous publication [2], our group reported that some of the complete blood count index-based calculated biomarkers had different predictive powers at various stages of the disease. With the same in mind, in the current study, we investigated how PIVs and SII and PNI values change with the course of CRC.

The PIV is the youngest of the three biomarkers and was first described in CRC [8]. It was developed with the aim of providing indirect but sufficiently accurate information on both the inflammatory and immune processes that occur alongside tumorous diseases, and of integrating several already known pro-inflammatory, tumor microenvironment-associated, and/or immune-inflammatory markers [8,27]. Since then, its predictive capabilities have been demonstrated in a few other cancerous and non-cancerous diseases [28,29,30]. In CRC, a high vs. low PIV is associated with a two-times higher OS and progression-free survival (PFS) [31,32]. The PIV is a good predictor of pre-, peri-, and postoperative indicators [33,34,35], but some conflicting results have also been published [36]. In relation to the clinicopathological parameters of CRC, it was reported previously that the PIV was a good marker that aligned with clinical stage [37], and age ≥ 65 years, distant metastases, KRAS wild type, and deficiency in DNA mismatch repair were associated with higher PIVs [38]. The association with distant metastases and clinical staging was described in the current study as well. A novel observation from our results is that, to our knowledge, no previous study has investigated the longitudinal change in PIV in such detail. It was described that the PIVs of surviving patients did not change with the course of the disease, while the PIVs of deceased patients were significantly higher. Moreover, an early PIV increase was more characteristic for those patients who died due to CRC-related causes, and a later PIV increase was more frequent for those who died due to non-cancerous reasons. In the research of Pérez-Martelo et al. [39] and Corti et al. [40], the authors found that the introduction of chemotherapy or immunotherapy reduces, while disease progression increases the PIV [39,40], which was also observed in the current study. We identified significantly lower PIVs in female CRC patients throughout the whole study, which is in line with the study by Seo et al. [35], but is in contrast with most of the previous studies [8,28,29,37,39,40]. We hypothesize that the constant lower PIVs of females that we observed might have occurred due to the known sexual dimorphism associated with CRC, including but not limited to the protective effect of estrogen on systemic inflammation and the gut microbiome [41]. In addition to all of the above, we found that none of the investigated comorbidities or relevant medical history events had any effect on CRC patients’ PIVs.

Similarly to PIV, SII is also an indirect immune-inflammatory biomarker, which is also found to be highly associated with OS, PFS, recurrence-free survival, and other survival indicators [42,43,44,45], independent of the anti-cancer treatment the patients receive [46,47,48]. An increase in SII following tumor removal surgery is associated with worse survival prognosis [49,50] and with more frequent postoperative complications [51,52,53]. Higher SII levels are associated with higher TNM stages, higher tumor grades, sarcopenia, KRAS mutations, lymphovascular invasion, older age, and right-sided CRC [38,44,54,55,56,57,58,59]. It is also suggested that the SII can be predictive in the early detection of CRCs or can even distinguish between CRC and adenomatous polyps [58,60,61]. In the current study, the same situation could be described for SII as for PIV, detailed above. It showed the same changes in the survivor, tumor, and other cause of death cohorts as the PIV, and higher SII values were associated with more advanced tumor stages and with rectal cancer. The dynamic change undergoing/following chemotherapy was in line with a previous study’s results [62]. No relationship with any of the comorbidities and/or medical history events investigated were identified.

PNI has been investigated in various cancerous diseases, including CRC, since the 1980s [63]. It was described early in the literature that the chronic systemic inflammation caused by cancer negatively affects the nutritional status of the patients [64]; therefore, PNI can also be used as a biomarker that can give indirect information about both the inflammatory and the nutritional status of cancer patients. Similarly to the previous two immune-inflammatory indices, a pathological and/or lower PNI value is known to be associated with worse survival outcomes [11,65,66,67,68], with significantly more postoperative complications [69], and with older age and worst clinicopathological parameters [67,68,70]. These observations could be repeated in the current study as well. Moreover, we found that CRC survivors had a very slow decrease in PNI value, similar to what is known to occur in healthy people [71]. In contrast, the PNI values decreased significantly in those patients who died throughout our study. The decrease was more intense in the first part of our observational period when the death was due to CRC, and a much smaller and later decrease was found in those who died due to non-cancerous causes. We also found that those CRC patients who had co-existing hypertension had constantly lower PNI levels throughout the study. PNI and hypertension have a known relationship [72,73], and we hypothesize that the observation we described might occur due to the following process: It is known that CRC can often be associated with hypoalbuminemia [74] and lymphopenia [75]. Both CRC and hypertension are known to cause chronic systemic inflammation, which can result in an inadequate malnutritional status [76]. This state is often undiagnosed, making it a silent but serious health risk, especially in people with chronic diseases [77]. And when the two diseases co-exist, these self-exciting processes can create a vicious cycle that eventually leads to even further decline, which can also be influenced by the further negative effects of chemotherapy.

### Limitations of This Study

This study has some limitations, starting with the retrospective design and the fact that the data was obtained from a single center only, which limits generalizability and might have introduced biases regarding the selection of cases, incomplete data, and limited control over confounders. The retrospective design has influenced our ability to further analyze the effects of other clinical parameters and their relationship with the PIV, SII, and PNI. Additional limiting factors were as follows: (1.) Not every laboratory measurement from the patients could be downloaded from the medical system, only randomly selected ones. (2.) Due to certain infrastructure and funding changes, not all laboratory measurements were available for all visits. To reduce biases connected to the latter two factors, we chose statistical methods that could robustly address the problem of missing values. (3.) Data on surgical details (type of operation, Clavien Dindo Classification categories, etc.) could not be collected, as a large number of patients were operated on at another institute. (4.) Although the literature data suggests that poor differentiation is significantly associated with systemic inflammation and high SII values [43,78], no data on histologic grading was collected.

## 5. Conclusions

In summary, our findings revealed that the PIV, SII, and PNI are potentially good indicators of patient survival in CRC at any given point throughout the course of the disease. We believe that the future use of the PIV, SII, and PNI would fit into the current treatment structure in a way that would complement rather than replace any other traditional routine markers, such as CEA, CA19-9, etc. It has to be highlighted, however, that due to the limitations of our study, such as the single-center retrospective design, the results presented are not fully generalizable; therefore, their use in everyday oncological practice should be further studied, possibly in a prospective multi-center study. There is a complex relationship between the various complete blood count indices, immune-inflammatory biomarkers, and CRC, and the tendencies indicated by the PIV, SII, and PNI suggest that further parameters and/or their combination, such as the measurement of macrophages and myeloid-derived suppressor cells, should be investigated as well. Our finding that PNI, CRC, and hypertension have a strong association warrants that cardiometabolic and/or nutritional adjustments are necessary prior any oncological interventions, possibly as a new oncological algorithm. Overall, we recommend the development of complex oncological treatment programs involving other specialists, which will ultimately help to detect changes in the disease state as early as possible.

## Figures and Tables

**Figure 1 medsci-13-00108-f001:**
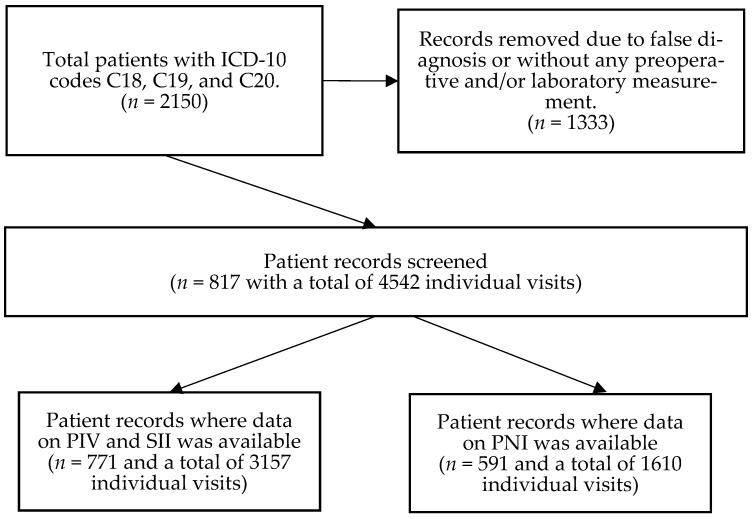
After the removal of ineligible patients, a total of 817 colorectal cancer patients were identified, of whom 771 and 591 patients had at least one pan-immune inflammation (PIV) and systemic immune-inflammation index (SII) value, and prognostic nutritional index (PNI) value, respectively.

**Figure 2 medsci-13-00108-f002:**
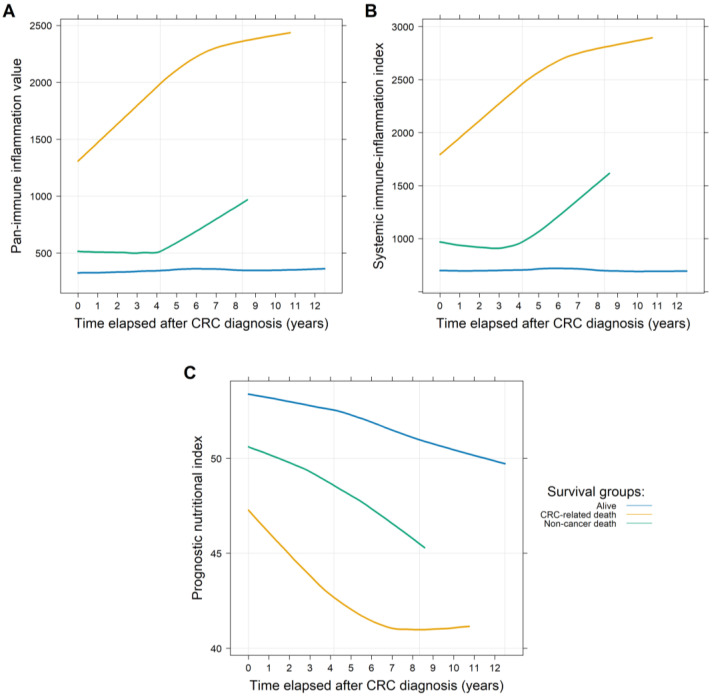
Longitudinal changes in the pan-immune inflammation (PIV; (**A**)), the systemic immune-inflammation index (SII; (**B**)), and the prognostic nutritional index (PNI; (**C**)) values throughout the observational period of the study. Higher PIVs and SII values and lower PNI values are associated with worse clinical outcomes. Those patients who died due to colorectal cancer (CRC) had significantly higher PIVs and SII values and lower PNI values throughout the entire study. Those patients who died due to other reasons or were still alive at the end of the observational period had more constant and similar values, compared to those who died due to CRC.

**Figure 3 medsci-13-00108-f003:**
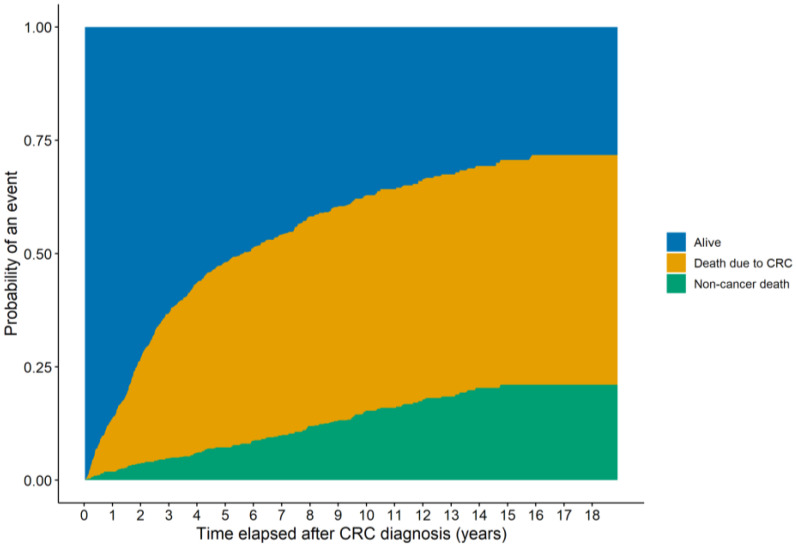
Cumulative incidences of two competing events throughout the observational period of this study. It was found that death due to colorectal cancer (CRC) occurred more frequently in the earlier part of the study, while death due to other causes was more frequent in the later period of the study.

**Figure 4 medsci-13-00108-f004:**
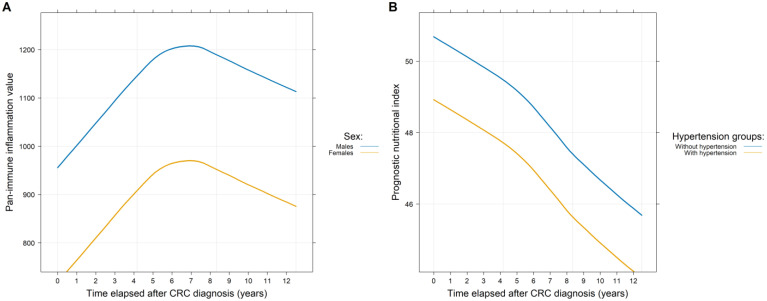
Throughout the whole observation period of the study, there was a significant difference between the pan-immune inflammation (**A**) and the prognostic nutritional index (**B**) values of the two sexes and those patients with and without hypertension, respectively.

**Figure 5 medsci-13-00108-f005:**
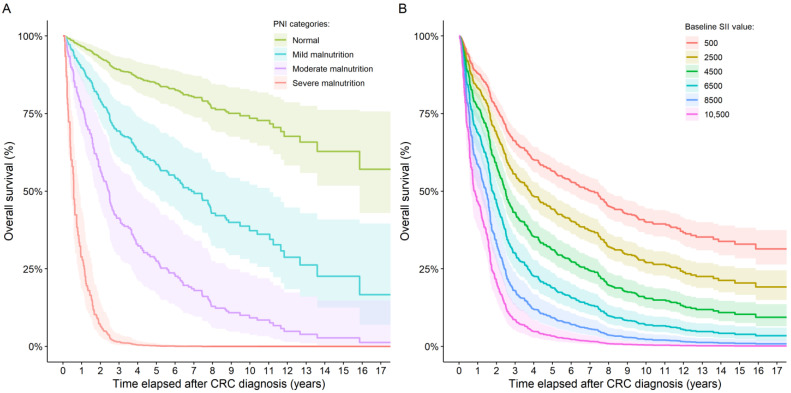
Overall survival curves for (**A**) patients with different SII values and (**B**) patients in the four PNI categories. No group number could be given for the PNI categories because dynamic group switching between the four groups was allowed in the model. The SII values shown in the figure are hypothetical values only.

**Table 1 medsci-13-00108-t001:** Complete anamnestic data of the study cohorts. Continuous, count, and survival data is presented as mean ± standard deviation, number of observations (percentage), and median survival time (95% confidence interval), respectively.

Parameter	Complete Cohort (*n* = 817)	Patients with PIV/SII Data (*n* = 771)	Patients with PNI Data (*n* = 591)
Age (years)	65.27 ± 10.97	65.29 ± 10.96	65.26 ± 11.08
Sex (male/female)	456:361 (55.81%:44.19%)	426:345 (55.25%:44.75%)	316:275 (53.47%:46.53%)
Primary tumor location			
- Left-sided	266 (32.56%)	250 (32.43%)	185 (31.30%)
- Right-sided	239 (29.25%)	232 (30.09%)	183 (30.96%)
- Rectal	293 (35.86%)	272 (35.28%)	209 (35.36%)
- Multiple sites	19 (2.33%)	17 (2.20%)	14 (2.37%)
AJCC stage (I/II/III/IV)	102:209:182:324 (12.48%:25.58%:22.28%:39.66%)	96:204:176:295 (12.45%:26.46%:22.83%:38.26%)	66:159:133:233 (11.17%:26.90%:22.50%:39.42%)
Regional lymph node metastasis	438 (53.61%)	408 (52.92%)	315 (53.50%)
Distant metastasis			
- Synchronous	323 (39.53%)	294 (38.13%)	232 (39.26%)
- Metachronous	98 (12.00%)	97 (12.58%)	75 (12.69%)
Inoperable cases	67 (8.20%)	47 (6.10%)	35 (5.92%)
Chemotherapy			
- None	248 (30.35%)	233 (30.22%)	172 (29.10%)
- Adjuvant	244 (29.87%)	238 (30.87%)	178 (30.12%)
- Palliative			
○ First-line	124 (15.18%)	110 (14.27%)	90 (15.23%)
○ Second-line	96 (11.75%)	87 (11.28%)	70 (11.84%)
○ Third-line or above	105 (12.85%)	103 (13.36%)	81 (13.71%)
Use of biological agents	224 (27.42%)	212 (27.50%)	173 (29.27%)
Use of regorafenib/trifluridine–tipiracil	53 (6.49%)	53 (6.87%)	45 (7.61%)
Patient with any relevant medical history and/or comorbidity	666 (81.52%)	627 (81.32%)	490 (82.91%)
Medical history			
- Type 2 diabetes mellitus	204 (24.97%)	193 (25.03%)	144 (24.37%)
- Thyroid disease ^1^	82 (10.04%)	77 (9.99%)	61 (10.32%)
- Hypertension (untreated/treated)	58:519 (7.10%:63.53%)	52:490 (6.74%:63.55%)	37:394 (6.26%:66.67%)
- Major CV events ^2^	87 (10.65%)	83 (10.77%)	65 (11.00%)
- Other CV diseases ^3^	64 (7.83%)	59 (7.65%)	52 (8.80%)
- Appendicitis with or without appendectomy	142 (17.38%)	136 (17.64%)	107 (18.10%)
- Cholelithiasis with or without cholecystectomy	191 (23.38%)	185 (23.99%)	143 (24.20%)
Median survival time (months)	68.04 (55.95–82.50)	58.78 (47.04–75.17)	59.76 (44.15–82.50)

^1^ In euthyroid state. ^2^ Myocardial infarction, stroke, coronary artery bypass grafting, and/or stent implantation. ^3^ Transient ischemic attack, (pulmonary) embolism, carotid endarterectomy, and/or other type of angioplasty. AJCC: American Joint Committee on Cancer; CV: cardiovascular; PIV: pan-immune inflammation value; PNI: prognostic nutritional index; SII: systemic immune-inflammation index.

**Table 2 medsci-13-00108-t002:** *p*-values obtained from univariate longitudinal mixed effect models investigating whether any comorbidities and/or clinicopathological parameters related to colorectal cancer affected the pan-immune inflammation (PIV), prognostic nutritional index (PNI), or the systemic immune-inflammation index (SII) values.

Parameter	PIV	SII	PNI
Age (years)	0.6798	0.2921	<0.0001
Sex	0.0286	0.2294	0.3724
Primary tumor location	0.1430	0.0268	0.6604
AJCC stage (I–III vs. IV)	<0.0001	<0.0001	<0.0001
Regional lymph node metastasis	<0.0001	<0.0001	0.0021
Synchronous distant metastasis	<0.0001	<0.0001	<0.0001
Metachronous distant metastasis	0.0911	0.0396	0.0911
Chemotherapy (none vs. adjuvant/palliative)	<0.0001	<0.0001	<0.0001
Use of biological agents	0.0021	0.0383	0.1340
Use of regorafenib/trifluridine–tipiracil	0.3777	0.8311	0.7411
Patient with any relevant medical history and/or comorbidity	0.4740	0.5667	0.0005
Medical history			
- Type 2 diabetes mellitus	0.4600	0.4553	0.8570
- Thyroid disease ^1^	0.1218	0.2272	0.8027
- Hypertension	0.4442	0.6826	0.0076
- Major CV events ^2^	0.8685	0.4778	0.5689
- CV diseases ^3^ incl. major CV events but excluding hypertension	0.4587	0.8093	0.1523
- Appendicitis with or without appendectomy	0.4970	0.7622	0.7221
- Cholelithiasis with or without cholecystectomy	0.5974	0.3869	0.5124

^1^ In euthyroid state. ^2^ Myocardial infarction, stroke, coronary artery bypass grafting, and/or stent implantation. ^3^ Transient ischemic attack, (pulmonary) embolism, carotid endarterectomy, and/or other type of angioplasty. AJCC: American Joint Committee on Cancer; CV: cardiovascular.

**Table 3 medsci-13-00108-t003:** *p*-value results of three multivariate survival models, where the three study targets were combined with sex and data on comorbidities and medical history.

Parameter	Model 1	Model 2	Model 3
Pan-immune inflammation value (increase per unit)	<0.0001	–	–
Systemic immune-inflammation index (increase per unit)	–	<0.0001	–
Prognostic nutritional index (increase per unit)	–	–	<0.0001
Sex [male (ref.) vs. female]	0.1296	0.0057	0.1624
Medical history			
- Type 2 diabetes mellitus [no (ref.) vs. yes]	0.3277	0.2530	0.2001
- Thyroid disease ^1^ [no (ref.) vs. yes]	0.4635	0.3910	0.1672
- Hypertension [no (ref.) vs. yes]			
○ None (ref.) vs. untreated hypertension	0.3049	0.5455	0.5521
○ None (ref.) vs. treated hypertension	0.0092	0.0754	0.0860
- CV diseases ^2^ incl. major CV events ^3^ but excluding hypertension [no (ref.) vs. yes]	0.9813	0.2624	0.3729
- Appendicitis with or without appendectomy [no (ref.) vs. yes]	0.6518	0.2197	0.1576
- Cholelithiasis with or without cholecystectomy [no (ref.) vs. yes]	0.7488	0.6521	0.1201

^1^ In euthyroid state. ^2^ Transient ischemic attack, (pulmonary) embolism, carotid endarterectomy, and/or other type of angioplasty. ^3^ Myocardial infarction, stroke, coronary artery bypass grafting, and/or stent implantation. CV: cardiovascular; ref.: reference category.

**Table 4 medsci-13-00108-t004:** *p*-value results of three multivariate survival models, where the three study targets were combined with the most commonly used colorectal-cancer-related clinicopathological parameters and a binarized version of data on comorbidities and medical history.

Parameter	Model 1	Model 2	Model 3
Pan-immune inflammation value (increase per unit)	<0.0001	–	–
Systemic immune-inflammation index (increase per unit)	–	<0.0001	–
Prognostic nutritional index (increase per unit)	–	–	<0.0001
Age (years)	<0.0001	0.0002	0.8240
Sex [male (ref.) vs. female]	0.8328	0.3808	0.1842
Patient with any relevant medical history and/or comorbidities [no (ref.) vs. yes]	0.7084	0.9684	0.0488
Primary tumor location			
- Left-sided (ref.) vs. right-sided	0.9618	0.9426	0.0572
- Left-sided (ref.) vs. rectum	0.5043	0.1921	0.0096
- Left-sided (ref.) vs. multiplex	0.0901	0.0697	0.7061
- Right-sided (ref.) vs. rectum	0.5031	0.2367	0.9482
- Right-sided (ref.) vs. multiplex	0.0888	0.0703	0.2147
- Rectum (ref.) vs. multiplex	0.1193	0.1260	0.1709
AJCC stage			
- Stage I (ref.) vs. stage II	0.2199	0.2001	0.1467
- Stage I (ref.) vs. stage III	0.0048	0.0039	0.0239
- Stage I (ref.) vs. stage IV	<0.0001	<0.0001	0.0002
- Stage II (ref.) vs. stage III	0.0214	0.1882	0.0327
- Stage II (ref.) vs. stage IV	<0.0001	<0.0001	<0.0001
- Stage III (ref.) vs. stage IV	<0.0001	<0.0001	<0.0001

AJCC: American Joint Committee on Cancer; ref.: reference category.

## Data Availability

The datasets used and/or analyzed during the current study are available from the corresponding author on reasonable request.

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
