# Peer review of "Are Calculated Immune Markers with or Without Comorbidities Good Predictors of Colorectal Cancer Survival? The Results of a Longitudinal Study"

_medsci, 2025, doi:10.3390/medsci13030108_

Round 1

Reviewer 1 Report

Comments and Suggestions for Authors

Thank you for the article. Here are some points in which the article could be improved:

  1. Regarding the inclusion criteria, were the emergency cases (perforation-obstruction due to tumour etc…) included in the study? Can you specify this information in the materials and method section?
  2. While doing research regarding the sentences: ‘Similarly to the previous two immune inflammatory indexes, pathological and/or lower PNI value is known to be associated with worse survival outcomes , with significantly more postoperative complications , and with older age and worst clinicopathological parameters’  and ‘Similarly to the previous two immune inflammatory indexes, pathological and/or lower PNI value is known to be associated with worse survival outcomes , with significantly more postoperative complications , and with older age and worst clinicopathological parameters’ It is strongly recommended that the manuscript will strongly benefit from this article titled ‘The relationship between the Prognostic Nutritional Index and lymphovascular and perineural invasion of the tumor in patients diagnosed with gastric cancer, and its effect on overall survival’ doi: 1097/MD.0000000000040087

3. Please give detailed information regarding the operations of the patients-whether they are operated, if yes please state which operation type they underwent. Also, please add a seperate table summarizing or grouping the operations.

4. In addition, please state if there were any complications due to these operations, since the complications are one of the most important factors that affect overall survival. Please group them according to Clavien Dindo Classification.

5. On line 128, instead of the phrase 'dummy variables' please use 'binary variables'

Author Response

We thank Reviewer 1 for the positive feedback on our article. Our answers to the questions and critical comments are below:

Thank you for the article. Here are some points in which the article could be improved:

  1. Regarding the inclusion criteria, were the emergency cases (perforation-obstruction due to tumour etc…) included in the study? Can you specify this information in the materials and method section?

Thank you for your kind comment. No, all emergency cases were excluded from the study. The exclusion criteria in Methods were updated.

  1. While doing research regarding the sentences: ‘Similarly to the previous two immune inflammatory indexes, pathological and/or lower PNI value is known to be associated with worse survival outcomes , with significantly more postoperative complications , and with older age and worst clinicopathological parameters’ and ‘Similarly to the previous two immune inflammatory indexes, pathological and/or lower PNI value is known to be associated with worse survival outcomes , with significantly more postoperative complications , and with older age and worst clinicopathological parameters’ It is strongly recommended that the manuscript will strongly benefit from this article titled ‘The relationship between the Prognostic Nutritional Index and lymphovascular and perineural invasion of the tumor in patients diagnosed with gastric cancer, and its effect on overall survival’ doi: 10.1097/MD.0000000000040087

Thank you for your suggestion. Although the proposed article is not closely related to the topic of the proposed section, we have managed to place it elsewhere.

  1. Please give detailed information regarding the operations of the patients-whether they are operated, if yes please state which operation type they underwent. Also, please add a seperate table summarizing or grouping the operations.

While data on oncological treatment and patient follow-up were available at our regional oncology center, surgical resection of primary tumors was performed not only at the surgical departments of Semmelweis University, but in other hospitals as well. For this reason, we have limited access to complete surgical data, such as operation type, operational time, etc. In general, we can report that depending on the location of the tumor, the most appropriate type of surgery was selected by a team of surgeons, oncologist and pathologists, according to national and ESMO guidelines. A short sentence has been included in the limitations.

However, some limited data were available in our database on irresectable cases. Table 1 has therefore been revised with this information. In addition, we found no difference in the comparisons between irresectable and resectable cases.

  1. In addition, please state if there were any complications due to these operations, since the complications are one of the most important factors that affect overall survival. Please group them according to Clavien Dindo Classification.

In line with our answer to question 3, Clavien Dindo Classification categories and data on surgical complications cannot be collected for at least half of the patients. It has to be mentioned though that all operated patients received their first oncological treatment at least 4-6 weeks after the surgery, on which visit the postoperative blood collection was also carried out. As the general good condition of the patient (absence of anemia, normal laboratory results, etc.) is a necessary condition for any oncological treatments, the timely initiation of these treatments was an indirect indicator of the patient's good condition without any serious complication related to the surgery.

  1. On line 128, instead of the phrase 'dummy variables' please use 'binary variables'

Thank you. It was corrected.

Reviewer 2 Report

Comments and Suggestions for Authors

I appreciate the authors for presenting a nice piece of research with potential implications for understanding and diagnosing the clinical manifestations of colorectal cancer. However, I would like to raise the following concerns for consideration:

  1. Objective Clarity is missing in the abstract. The background/objective section of the abstract (lines 14–16) does not clearly depict the study's objectives. The authors are advised to state the objectives more robustly to reflect the research questions being asked in this manuscript.
  2. In line 130, the abbreviation "OS" is used. Please clarify what "OS" stands for to ensure clarity for readers unfamiliar with the term. Is this a typo error or something else?
  3. The size of the sample applied in the study appears limited, and there is a lack of information regarding uniform diagnostic criteria across the samples. Further elaboration on these aspects would strengthen the study’s methodological rigor.
  4. The manuscript's coherence requires improvement. Specifically, although the title includes the term "comorbidity," the authors neither mention nor discuss it in the introduction. To enhance clarity and consistency, authors should revise the abstract and introduction to accurately reflect the study’s objectives and the research questions being addressed.

Author Response

We thank Reviewer 2 for the positive feedback on our article. Our answers to the critical questions and comments are below:

I appreciate the authors for presenting a nice piece of research with potential implications for understanding and diagnosing the clinical manifestations of colorectal cancer. However, I would like to raise the following concerns for consideration:

  1. Objective Clarity is missing in the abstract. The background/objective section of the abstract (lines 14–16) does not clearly depict the study's objectives. The authors are advised to state the objectives more robustly to reflect the research questions being asked in this manuscript.

We thank the Reviewer for their kind critical comment. The abstract of the manuscript was revised as follows: Background/Objectives: Although numerous prognostic biomarkers have been proposed for colorectal cancer (CRC), their longitudinal evaluation remains limited. Aims of the study was to investigate longitudinal changes of biomarkers calculated from routinely used laboratory markers, and their relationship to common chronic diseases (comorbidities).”

  1. In line 130, the abbreviation "OS" is used. Please clarify what "OS" stands for to ensure clarity for readers unfamiliar with the term. Is this a typo error or something else?

OS is the abbreviation of overall survival. The sentence was revised.

  1. The size of the sample applied in the study appears limited, and there is a lack of information regarding uniform diagnostic criteria across the samples. Further elaboration on these aspects would strengthen the study’s methodological rigor.

Thank you for pointing out this important issue. The following was added to Methods. “CRC was diagnosed using colonoscopy followed by the pathological confirmation of colorectal adenocarcinoma from biopsy specimens, and further imaging studies were performed to confirm metastases.

  1. The manuscript's coherence requires improvement. Specifically, although the title includes the term "comorbidity," the authors neither mention nor discuss it in the introduction. To enhance clarity and consistency, authors should revise the abstract and introduction to accurately reflect the study’s objectives and the research questions being addressed.

We thank the Reviewer for their kind critical comment and suggestion. Indeed, the original version of the Introduction lacked a paragraph about this relationship. To gain better clarity, a new paragraph about the association and relationship between CRC and comorbidities was added to the introduction of the manuscript. The abstract and the aims of the study were also edited.

Reviewer 3 Report

Comments and Suggestions for Authors

The manuscript presents a retrospective longitudinal study to delineate the modulations and associations of PVI, SII, and PNI with CRC survival.

Strengths:

  • The study is relevant, given the important role of inflammation in pathogenesis and metastatic progression of colorectal cancer, and the need for reliable biomarkers to determine the prognosis of the disease.
  • The experimental design is comprehensive, evaluating PVI, SII, and PNI in 817 colorectal cancer patients, including their histories and co-morbidities, analyzing 4542 measurement points.
  • The study has made a novel observation of PIV being significantly higher in the deceasing patients.
  • Robust statistical methods to address limitations.

Areas for Improvement:

  • All data comes retrospectively from a single institution in Hungary. The authors should discuss how local practice patterns or population characteristics may limit generalizability to other settings.
  • While the association between pathological changes in PIV, SII, and PNI and poor prognosis is statistically significant, causality cannot be inferred. The authors should discuss conclusions accordingly.

Author Response

We thank Reviewer 3 for the positive feedback on our article. Our answers to the critical comments are below:

The manuscript presents a retrospective longitudinal study to delineate the modulations and associations of PVI, SII, and PNI with CRC survival.

Strengths:

  • The study is relevant, given the important role of inflammation in pathogenesis and metastatic progression of colorectal cancer, and the need for reliable biomarkers to determine the prognosis of the disease.
  • The experimental design is comprehensive, evaluating PVI, SII, and PNI in 817 colorectal cancer patients, including their histories and co-morbidities, analyzing 4542 measurement points.
  • The study has made a novel observation of PIV being significantly higher in the deceasing patients.
  • Robust statistical methods to address limitations.

Areas for Improvement:

  1. All data comes retrospectively from a single institution in Hungary. The authors should discuss how local practice patterns or population characteristics may limit generalizability to other settings.

Thank you for your critical comment. The manuscript was revised to address these critical issues.

  1. While the association between pathological changes in PIV, SII, and PNI and poor prognosis is statistically significant, causality cannot be inferred. The authors should discuss conclusions accordingly.

We thank the Reviewer for their critical comment. The manuscript was revised accordingly.

Reviewer 4 Report

Comments and Suggestions for Authors

Summary Statement

 In their paper entitled “Are Calculated Immune Markers with or without Comorbidities Good Predictors of Colorectal Cancer Survival? Results of a Longitudinal Study.”, Herold et al investigate pan-immune inflammation value (PIV), prognostic nutritional index (PNI), and systemic immune-inflammation index (SII) as prognostic markers for colorectal cancer survival, overall survival, and possible early detection of changes in disease state. Overall, the study and paper are well done; however, several potential limitations of the study need to be addressed.

Comments

-What exactly does colorectal cancer in your paper stand for? Is this just conventional adenocarcinoma? There are a plethora of other colon cancer types (neuroendocrine, hepatoid, adenosquamous, lymphoglandular complex-like colorectal carcinoma, etc.). The treatment differs for most of these, so a clarifying statement if you are prototypically talking about adenocarcinoma needs to be made so readers know the extent of generalizability of your work.

-Your exclusion criteria do not include hepatic disorders or renal disease? Is there a reason for this? Hepatic function is necessary for albumin production, which is used in the calculation of PNI. Thrombopoietin is also made by the liver, a reduction of which is known to cause thrombocytopenia and would thus lower SII. Ideally, you would control for this variable; however, a statement should at least be made of this limitation.

-Your exclusion criteria do not include renal dysfunction. Hypoalbuminemia due to albuminuria is associated with some renal disorders. In some other disorders, increases in albumin in serum can be seen due to a lack of clearance. A statement should be made of this limitation if it is not explicitly controlled for. Controlling for hypertension and type 2 diabetes mellitus is not enough to eliminate this bias.

-How do PIV, SII, and PNI compare to the current use of biomarkers such as CEA, CA19.9, cell-free DNA, etc? A statement about this would add generalizability and gain broader reader attention.

-In your analysis, histologic grading of the carcinoma is not addressed. Is there a reason for this? Well-differentiated tumors usually cause less systemic inflammation than poorly differentiated tumors.

Author Response

We thank Reviewer 4 for your work and time. Here we provide answers to the questions raised by the Reviewer:

Summary Statement

 In their paper entitled “Are Calculated Immune Markers with or without Comorbidities Good Predictors of Colorectal Cancer Survival? Results of a Longitudinal Study.”, Herold et al investigate pan-immune inflammation value (PIV), prognostic nutritional index (PNI), and systemic immune-inflammation index (SII) as prognostic markers for colorectal cancer survival, overall survival, and possible early detection of changes in disease state. Overall, the study and paper are well done; however, several potential limitations of the study need to be addressed.

Comments

  1. What exactly does colorectal cancer in your paper stand for? Is this just conventional adenocarcinoma? There are a plethora of other colon cancer types (neuroendocrine, hepatoid, adenosquamous, lymphoglandular complex-like colorectal carcinoma, etc.). The treatment differs for most of these, so a clarifying statement if you are prototypically talking about adenocarcinoma needs to be made so readers know the extent of generalizability of your work.

Thank you for your critical comment. Indeed, all study participants had conventional adenocarcinoma, and other types were excluded from the study. The Methods section of the manuscript was revised accordingly.

  1. Your exclusion criteria do not include hepatic disorders or renal disease? Is there a reason for this? Hepatic function is necessary for albumin production, which is used in the calculation of PNI. Thrombopoietin is also made by the liver, a reduction of which is known to cause thrombocytopenia and would thus lower SII. Ideally, you would control for this variable; however, a statement should at least be made of this limitation.

Thank you for pointing out this important issue. All known hepatic disorder(s) except for Gilbert’s syndrome were also an exclusion criterion of the study. Exclusion criteria included potentially all conditions that are known to infer with complete blood count parameters. The corresponding text in Methods was revised accordingly. The determination of thrombopoietin level is not part of the routine care in Hungary.

  1. Your exclusion criteria do not include renal dysfunction. Hypoalbuminemia due to albuminuria is associated with some renal disorders. In some other disorders, increases in albumin in serum can be seen due to a lack of clearance. A statement should be made of this limitation if it is not explicitly controlled for. Controlling for hypertension and type 2 diabetes mellitus is not enough to eliminate this bias.

Thank you for pointing out this important issue. Renal dysfunction was also an exclusion criterion. eGFR was manually calculated for every single measurement using the CKD-EPI equation (data not presented in the manuscript). The eGFR of the patients with available PIV/SII and PNI at the time of CRC diagnosis was 79.46 ± 19.40 ml/min/1.73m2 (mean ± SD) and 79.39 ± 19.35 ml/min/1.73m2, respectively. Exclusion criteria in Methods were revised.

  1. How do PIV, SII, and PNI compare to the current use of biomarkers such as CEA, CA19.9, cell-free DNA, etc? A statement about this would add generalizability and gain broader reader attention.

We hypothesize that PIV, SII and PNI can completement rather than replace the currently used routine markers. The following statement was added to Conclusion. “We believe that the future use of PIV, SII and PNI would fit into the current treatment structure in a way that they would complement rather than replace any other traditional routine markers such as CEA, CA19-9, etc.

  1. In your analysis, histologic grading of the carcinoma is not addressed. Is there a reason for this? Well-differentiated tumors usually cause less systemic inflammation than poorly differentiated tumors.

The Reviewer draws attention to a very important point. Unfortunately, when building our database, histological grading was not recorded. Although we fully agree on the importance of this issue, at this stage of the study, it is unfortunately not possible to collect the data in a short time and integrate it into the publication. In the future, we will certainly add this data to our database. The following was added to the Limitations of the study section: “Although literature data suggests that poor differentiation is significantly associated with systemic inflammation and high SII [PMID: 31496619 and 39806457], no data on histologic grading was collected.”

Round 2

Reviewer 1 Report

Comments and Suggestions for Authors

Thank you for the revisions. The current revised version of the manuscript  has been thoroughly revised and is now suitable for publication

Reviewer 2 Report

Comments and Suggestions for Authors

Authors,

Your explanations have adequately resolved my queries, and I find your revisions satisfactory. I have no further comments, and I am good with the acceptance of your updated manuscript.

With best

Abhishek Kumar

Reviewer 4 Report

Comments and Suggestions for Authors

The authors have addressed all of my concerns and the paper is ready for publication.